# Monkeypox: A Histopathological and Transmission Electron Microscopy Study

**DOI:** 10.3390/microorganisms11071781

**Published:** 2023-07-09

**Authors:** Chiara Moltrasio, Francesca Laura Boggio, Maurizio Romagnuolo, Rachele Cagliani, Manuela Sironi, Alessandra Di Benedetto, Angelo Valerio Marzano, Biagio Eugenio Leone, Barbara Vergani

**Affiliations:** 1Dermatology Unit, Fondazione IRCCS Ca’ Granda Ospedale Maggiore Policlinico, 20122 Milan, Italy; maurizio.romagnuolo@unimi.it (M.R.); alessandra.dibenedetto@policlinico.mi.it (A.D.B.); angelo.marzano@unimi.it (A.V.M.); 2Pathology Unit, Fondazione IRCCS Ca’ Granda Ospedale Maggiore Policlinico, 20122 Milan, Italy; francesca.boggio@policlinico.mi.it; 3Department of Pathophysiology and Transplantation, Università Degli Studi di Milano, 20122 Milan, Italy; 4Scientific Institute IRCCS E. MEDEA, Bioinformatics, 23842 Bosisio Parini, Italy; rachele.cagliani@lanostrafamiglia.it (R.C.); manuela.sironi@lanostrafamiglia.it (M.S.); 5Department of Medicine and Surgery, University of Milano-Bicocca, 20900 Monza, Italy; biagioeugenio.leone@unimib.it (B.E.L.); barbara.vergani@unimib.it (B.V.)

**Keywords:** monkeypox, virology, cutaneous manifestations, histopathology, transmission electron microscopy

## Abstract

The global outbreak of human monkeypox virus (hMPXV1) in 2022 highlighted the usefulness of dermatological manifestations for its diagnosis. Infection by the human monkeypox virus thus necessitated inclusion in the diagnostic repertoire of dermatopathology. To assess the histopathological and microscopical findings of cutaneous lesions related to hMPXV infection, we analyzed skin biopsies from patients with positive MPXV DNA polymerase chain reaction presenting with a typical course of hMPXV1 infection. The most prominent histopathological findings were ascribable to a pustular stage in which epidermal necrosis with areas of non-viable keratinocytes and a “shadow cell” appearance were evident; in some cases, the deep portion of the hair follicle and the acrosyringial epithelium were affected. The main cytopathic modifications included ballooning keratinocytes, followed by Guarnieri bodies and a ground glass appearance of the keratinocytes’ nuclei, together with a dense mixed inflammatory cell infiltrate with prominent neutrophil exocytosis. Transmission electron microscopy analysis demonstrated viral particle aggregates in the cytoplasm of keratinocytes, without any involvement of the nucleus. Interestingly, we also found the presence of viral particles in infected mesenchymal cells, although to a lesser extent than in epithelial cells. Through this study, we contributed to expanding the histological and microscopic knowledge of the human mpox virus, a key step to understanding current and potential future trends of the disease, as well as of other *Orthopoxvirus* infections.

## 1. Introduction

Mpox is a zoonotic disease caused by the monkeypox virus (MPXV), which belongs to the *Orthopoxvirus* genus in the *Poxviridae* family. In 1970, the first case of human MPXV infection was reported in the Democratic Republic of the Congo. Subsequently, the virus became endemic in Central and West Africa, where two distinct clades are prevalent: the Congo Basin clade (clade I) and the West African clade (clade II) [1].

Since May 2022, an unprecedented mpox outbreak has been reported in multiple non-endemic countries, especially in the United States and Europe. Consequently, MPXV is considered the most important human orthopoxviral infection since the eradication of smallpox in 1977 [2].

In the 2022 MPXV outbreak, most cases occurred among men who have sex with men (MSM), suggesting an early cluster, since the virus may spread through interconnected sexual networks [3]. The incubation period ranged from 5 to 21 days. Systemic symptoms included, among others, fever, headache, and asthenia. Within 1 to 5 days after the onset of fever, there was the appearance of cutaneous lesions that typically progressed from papules to vesiculopustules with central crusted umbilication. A range of complications such as secondary bacterial infections, gastrointestinal involvement, respiratory distress, and encephalitis were also reported [4]. The main diagnostic procedures consist of culture-based isolation and/or polymerase chain reaction (PCR) for MPXV DNA. However, serologic testing, histopathological examination, and transmission electron microscopy (TEM) can also be considered useful tools to confirm the diagnosis [5,6].

The World Health Organization lists different conditions, including, chickenpox, vaccinia, and herpes simplex, as communicable diseases to be differentiated from mpox [7]. In the clinical setting, a prompt diagnosis is fundamental and can be achieved by performing a PCR test from skin lesion swabs. However, if PCR is not available or if the clinical picture is not straightforward, histopathological findings allow for a presumptive diagnosis of mpox. An aspect to be considered is that the cutaneous histopathological changes occurring in mpox may resemble those attributable to other *Orthopoxviruses* [8]. Nevertheless, as demonstrated by the first reported case [9], histopathological examination of biopsy specimens may still play a fundamental role in the diagnosis of hMPXV infection.

Likewise, TEM studies focused on viral particle morphology and viral replication may also contribute to the diagnosis, while also shedding light on novel viral pathophysiological mechanisms, which could be useful for therapeutic purposes.

To date, despite the public health relevance of mpox, histopathological features of cutaneous lesions of hMPXV infection have not been extensively described, and TEM images of MPXV particles have rarely been reported. Here, we present a detailed clinical, histopathological, and TEM description of cutaneous lesions caused by hMPXV infection.

## 2. Materials and Methods

### 2.1. Patients

The study included six patients with MPXV infection attending the sexually transmitted diseases outpatient service of the Dermatology Unit of Fondazione IRCCS Ca’ Granda Ospedale Maggiore Policlinico, Milan, Italy, from June to July 2022 [10,11,12]. 

Positivity for MPXV DNA was confirmed through real-time polymerase chain reaction (RT-PCR) (RTPCR018-LPD-R, Vircell Microbiologists) on both pharyngeal and vesico-pustular fluid swabs for all cases. In one patient, the lesional swab also tested positive for *Treponema pallidum* (TP)—the microaerophilic spirochete responsible for syphilis—as well as in another patient, in whom serological tests were consistent with late latent syphilis.

All procedures were in accordance with the ethical standards of the Helsinki Declaration, and all patients provided written informed consent to be included in the study. Additionally, each patient signed a specific written informed consent for publication. All patient details were anonymized as much as possible to ensure the privacy and confidentiality of the patients. 

The study was approved by Fondazione IRCCS Ca’ Granda Ospedale Maggiore Policlinico of Milan, Italy (RC_2022/2023). 

### 2.2. Histology and Transmission Electron Microscopy

The skin specimens were obtained via 4 mm punch biopsy. Subsequently, sections of each case were stained with hematoxylin and eosin, according to the standard protocol [13]. 

A transmission electron microscopy study of the skin samples was also performed. Skin biopsies were fixed using a primary glutaraldehyde fixative solution (2% glutaraldehyde and 4% paraformaldehyde) in sodium cacodylate buffer. After post-fixation for 1 h in 2% osmium tetroxide and dehydration in ethanol, the samples were embedded in polybed 812 resin. Ultrathin sections (60–80 nm) were contrasted with uranyl acetate and lead citrate and observed with a Philipps CM120 electron microscope equipped with a Megaview III digital camera. 

## 3. Results

### 3.1. Clinical Features

The clinical features of our six patients are summarized in Table 1.

All patients were men who have sex with men (MSM) and revealed to have had unprotected sex in the weeks preceding MPXV infection. Two patients also presented with a syphilis co-infection. One patient traveled to the Netherlands in the three weeks prior to symptom onset, whereas another patient reported having traveled to Spain in the two weeks before the onset of cutaneous and systemic symptoms. The remaining patients had not traveled. Cutaneous manifestations mainly consisted of multiple pustules principally located on the penile shaft (n = 4) and pubic region (n = 3); the anal/perianal region (n = 2) and hands (n = 2) were also affected, as well as the upper (n = 1) and lower limbs (n = 1), face (n = 1), and perioral region (n = 1). All patients presented with systemic symptoms, including asthenia (n = 5), headache (n = 4), and fever (n = 3). One patient presented with hyperpyrexia and rectal pain two weeks prior to cutaneous manifestations, while another suffered from myalgia and arthralgia in addition to the other symptoms mentioned above. Laterocervical and inguinal lymphadenopathy was observed in two patients. Only one patient required hospitalization for logistical reasons of isolation. All patients achieved complete remission without any active medical treatment.

### 3.2. Histopathological Features 

The histopathological features are summarized in Table 2.

Most cases (n = 4) exhibited histological modification ascribable to a pustular stage [14]. In these cases, epidermal necrosis was evident, with areas of non-viable keratinocytes and a “shadow cell” appearance characterized by enlarged cells with eosinophilic cytoplasm, retained cell outline, but blurred or not evident nuclei (Figure 1). Meanwhile, the few viable elements displayed variable cytopathic modifications. In addition, in these four patients, mild to moderate perivascular and periadnexal inflammatory infiltrate with neutrophils was admixed with epidermal modifications (Figure 2).

In two out of six skin specimens, degenerative epithelial alterations in the acrosyringial epithelium were also present.

Variable follicular involvement was evident in three patients, in which slight and focal dyskeratosis of the deep dermal portion of the hair follicle represented the main histological modification.

Only one skin biopsy was identified as being in the vesicular stage, with epidermis showing acanthosis, spongiosis, and ballooning degeneration. No evidence of necrosis or multinucleated elements was observed.

Two cases were also characterized by dense perivascular, interstitial and periadnexal lymphocytic infiltration.

Overall, regarding major cytopathic modifications, balloon cell degeneration of keratinocytes (Figure 3) was present in half of the cases. Guarnieri bodies, which are eosinophilic homogeneous intracytoplasmic inclusions (Figure 4), were seen in five patients. Keratinocytes showing eosinophilic “ground glass” nuclei appearance within the central area of the nucleus and chromatin margination giving a basophilic “halo” (Figure 5) were detected in two patients. Occasional multinucleated keratinocytes (Figure 3) were also present in one out of six patients.

Lastly, positive immunohistochemical staining for *Treponema pallidum* revealed the presence of spirochetes within the cytoplasm of keratinocytes (patients 4 and 6) with perivascular distribution (patient 6).

### 3.3. Transmission Electron Microscopy Findings

The electron microscopic findings are summarized in Table 3. 

Ultrastructural analysis performed by electron microscopy on histological samples recovered from paraffin blocks allowed us to observe, in five out of six cases, the presence of viral particles—often in aggregates—in the cytoplasm of keratinocytes, without any involvement of the nucleus (Figure 6). The morphology of the virions ranged from roundish particles without any core inside (immature forms) to oval membranous bodies containing biconcave or brick-shaped electron-dense central cores (mature viruses) (Figure 7).

Interestingly, in our cases, no viral particles were localized inside the stratum corneum and between the scales. Moreover, in two samples, virions were also identified inside the cytoplasm of mesenchymal cells (Figure 8), although in lower numbers than in epithelial cells.

## 4. Discussion

The first histopathological description of hMPXV infection was carried out by Stagles et al. [15]. In their patients, prominent features included epidermal necrosis, multinucleated giant keratinocytes, keratinocytes with prominent nucleoli, and Guarnieri bodies accompanied by dermal edema and perivascular inflammatory cell infiltrate. Subsequently, Bayer-Garner [14] performed an accurate histopathological, immunohistochemical, and electron microscopic study on three skin specimens obtained from two patients with PCR positivity for MPXV. Typical findings included multinucleated syncytial keratinocytes, keratinocytes with ballooned nuclei with an eosinophilic “ground glass” appearance, spongiosis, and acanthosis—with few viable keratinocytes—progressing to full thickness necrosis. In addition, a lichenoid-mixed infiltrate with progressive exocytosis was present; a follicular, perivascular, and perieccrine inflammatory cell infiltrate was also described. Finally, electron microscopy revealed virions at various stages of assembly—mature and immature brick-shaped virions—within the keratinocyte cytoplasm. 

During the 2022 mpox outbreak, very few studies reported histopathological, immunohistochemical, and electron microscopic features related to hMPXV infection. Rodriguez-Cuadrado et al. [16] described clinical, histopathological, immunohistochemical, and electron microscopic findings in mpox cutaneous lesions, revealing epidermal necrosis and ballooning keratinocytes as the most recurrent histopathological characteristics. Notably, the cytoplasm of these keratinocytes showed strong positivity with the anti-*Vaccinia virus* antibody. In the same study, electron microscopy was performed in four cases, demonstrating numerous mpox viral particles in infected keratinocytes. 

More recently, Ortins-Pina et al. [9] described two mpox cases, in which the main histopathological findings consisted of epidermal necrosis, ballooning and reticular degeneration of keratinocytes, multinucleated keratinocytes, cytoplasmic eosinophilic Guarnieri-type inclusions, and a dense mixed inflammatory cell infiltrate with remarkable neutrophil exocytosis.

In line with previous observations, in five out of our six human mpox cases, we found that the most recurrent histopathological findings consisted of balloon cell degeneration of keratinocytes, followed by Guarnieri bodies, ground glass appearance of the keratinocytes’ nuclei, and multinucleated keratinocytes. Most cases presented with epidermal necrosis, accompanied by a modest follicular involvement characterized by a focal dyskeratosis of the deep portion of the hair follicle. The dermis was mainly characterized by moderate perivascular and periadnexal inflammatory cell infiltrate, primarily composed of neutrophils. However, in one case, the inflammatory cell infiltrate consisted only of lymphocytes. In one patient with syphilis co-infection, as confirmed by immunohistochemical positive staining for *Treponema pallidum*, we failed to find the previously mentioned histopathological features closely related to mpox infection; we observed a focal follicular dyskeratosis accompanied by severe periadnexal and interstitial lymphocytic infiltrate with numerous plasma cells. Similarly, from an ultrastructural point of view, we did not detect the virus, and this may be attributable to the timing of the biopsy.

In another patient affected by late latent syphilis, immunohistochemical staining for TP showed the presence of spirochetes in the cytoplasm of degenerated keratinocytes. This finding leads to the hypothesis that TP colonization could represent a consequence of an inflammatory memory recall determined by MPXV [17]. 

Overall, all reported cases, including those detailed in our study, displayed (i) acanthosis and necrosis of keratinocytes, ranging from single cell necrosis to full thickness epidermal necrosis; (ii) neutrophilic exocytosis that characterizes the pustular stage; and (iii) keratinocytes having ballooned nuclei with a ground glass appearance and multinucleated keratinocytes. The Guarnieri bodies, although not consistently found, are the hallmark of a poxvirus infection with active viral replication [8,9,14,15,16]. 

An aspect to be considered is that the histopathological changes occurring in *Orthopoxvirus* infections are similar between different species [18,19]. However, *Herpesvirus* can be histologically differentiated from *Orthopoxvirus*: herpes simplex virus (HSV) and varicella zoster virus (VZV) show viral cytopathic modifications consisting of ballooning degeneration, “steel gray” appearance of the nuclei with chromatin margination, and eosinophilic nuclear inclusions surrounded by a clear “halo’’ [19]. In contrast, eosinophilic inclusion bodies in *Orthopoxvirus* infections are detected in the cytoplasm of infected keratinocytes.

From an ultrastructural point of view, in five out of our six patients, we observed viral particles of different shapes in the cytoplasm of keratinocytes: roundish particles without any core inside representing immature forms, and mature viruses characterized by oval membranous bodies with biconcave or brick-shaped electron-dense central cores. These findings coincide with the observations reported by Bayer-Garner [14]. 

In two cases, mature and immature viruses were also detected, although at low abundance, in the cytoplasm of mesenchymal cells, as previously observed in studies conducted on a cynomolgus macaque model of aerosolized MPXV infection [20]. 

The presence of *Orthopoxviruses* in the small dermal blood vessels represents the beginning of skin infection with consequent development of specific cutaneous phenotypes, although the route through which the virus reaches the upper skin layers has not yet been clarified [21]. It has been postulated that infected Langerhans cells as migratory dendritic cells may play a role in this process by being susceptible to vaccinia virus (VACV) infection [20]. It is also important to consider that infection can also occur through the skin as the primary site; upon cutaneous inoculation, the targets of the virus are represented by keratinocytes and dermal fibroblasts, as well as Langerhans cells, dendritic cells, and macrophages. Subsequently, the virus can spread to distant organs or return to the skin through the lymphatic system [22]. 

To the best of our knowledge, this is the first study reported during the 2022 MPXV outbreak that demonstrates the presence of viral particles in infected mesenchymal cells. 

The main limitation of this study is the lack of immunohistochemical validation through anti-vaccinia virus antibodies, which produce a distinct and strong positive staining for ballooned and necrotic keratinocytes and provide direct evidence of MPXV infection. However, histopathology and TEM are key tools in the diagnosis of viral skin manifestations, allowing for rapid visualization of the virus, its morphology, and the different sites of its replication. Moreover, both the histology and TEM of MPXV infection, while showing similarities with other *Orthopoxviruses*, exhibit specific features that allow for a differential diagnosis, especially with *Herpesvirus* infections. In addition to the already mentioned differential diagnosis between *Orthopoxviruses* and *Herpesviruses* made possible by histology, a similar situation occurs in TEM study. Ultrastructurally, *Herpesviruses* differ from *Orthopoxviruses* by the presence of a dense core with a hexagonal capsid surrounded by a multilayer protein coat [23]; moreover, *Herpesviruses* replicate in the nucleus, as opposed to *Orthopoxviruses*, whose replication cycle occurs in the cytoplasm of infected cells [14]. 

Another limitation of this study is the small sample size, despite being one of the largest studies in terms of transmission electron microscopy examination during the 2022 mpox outbreak.

## 5. Conclusions

Since the start of the 2022 MPXV outbreak, 88,026 cases have been reported in total [24], with most cases managed in sexual health services. All recovered individuals are considered protected against reinfection, although anecdotal cases of individuals with apparent reinfection have recently been reported [25,26,27]. This implies greater knowledge not only in terms of treatment but also in terms of diagnosis. Undeniably, the role of molecular biology is fundamental for an accurate and rapid diagnosis. However, if PCR is not available or if the diagnosis is in doubt, skin biopsy and TEM may aid in the diagnostic process, thus improving clinical decisions and patient care. 

Based on our experience and literature review, the main histopathological findings in vesiculopustular lesions of mpox include ballooning keratinocytes, followed by Guarnieri bodies and ground glass appearance of the keratinocytes’ nuclei, together with a dense mixed inflammatory cell infiltrate with prominent neutrophil exocytosis. Although the histopathology of mpox lesions is similar to that of other *Orthopoxviruses*, histopathological examination may still allow for a mpox diagnosis in the presence of clinical suspicion and/or when molecular tests are not available. 

As far as TEM studies are concerned, they demonstrated viral particle aggregates ranging between 100 and 150 nm in size within the cytoplasm of infected keratinocytes, distinguishing mpox from other known members of the *Orthopoxvirus* genus. For the first time, during the 2022 MPXV outbreak, we demonstrated the presence of viral particles (both immature and mature forms) in infected mesenchymal cells, although to a lesser extent than in epithelial cells, thus expanding the microscopic knowledge of the human mpox virus and highlighting the importance of TEM for further investigation of this emerging pathogen.

In conclusion, the recent emergence of the mpox virus highlights the importance of including it in the diagnostic repertoire of dermatopathology. However, more research is needed to understand current and potential future trends of this disease, as well as of other *Orthopoxvirus* infections.

## Figures and Tables

**Figure 1 microorganisms-11-01781-f001:**
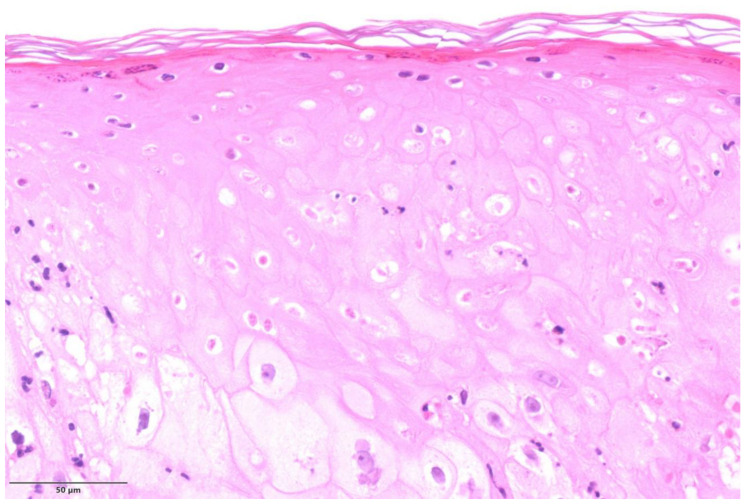
Non-viable keratinocytes and a “shadow cell” appearance characterized by enlarged cells with eosinophilic cytoplasm without well-defined nuclei (H&E, original magnification ×200).

**Figure 2 microorganisms-11-01781-f002:**
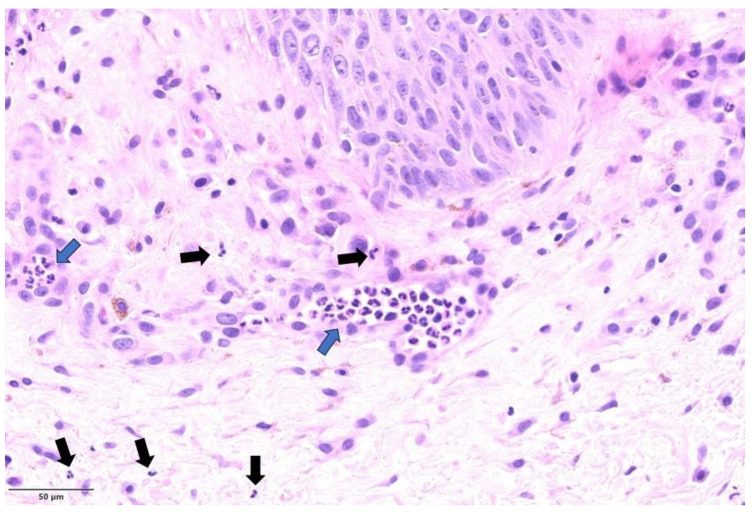
The mpox pustular stage characterized by dermal neutrophilic infiltration (black arrows), also present within the capillary vessels (blue arrow) (H&E, original magnification ×200).

**Figure 3 microorganisms-11-01781-f003:**
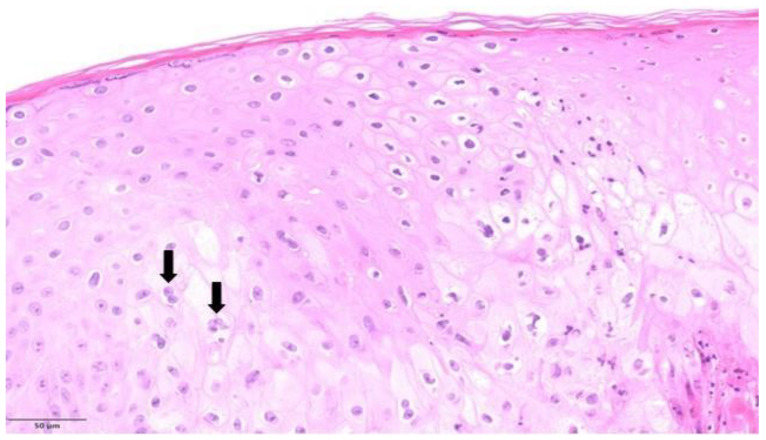
Balloon cell degeneration of keratinocytes. In ballooning degeneration, keratinocytes appear swollen, pale, and round due to intracellular edema and loss of intercellular bridges. Occasional multinucleated keratinocytes were also present (black arrows). (H&E, original magnification ×200).

**Figure 4 microorganisms-11-01781-f004:**
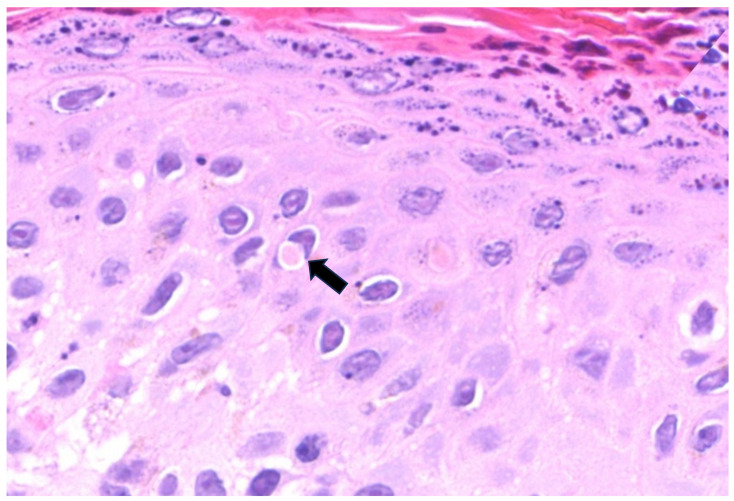
Ballooned, viable keratinocytes, one of which is harboring a cytoplasmic eosinophilic inclusion (arrow), also called a Guarnieri body (H&E, original magnification ×400).

**Figure 5 microorganisms-11-01781-f005:**
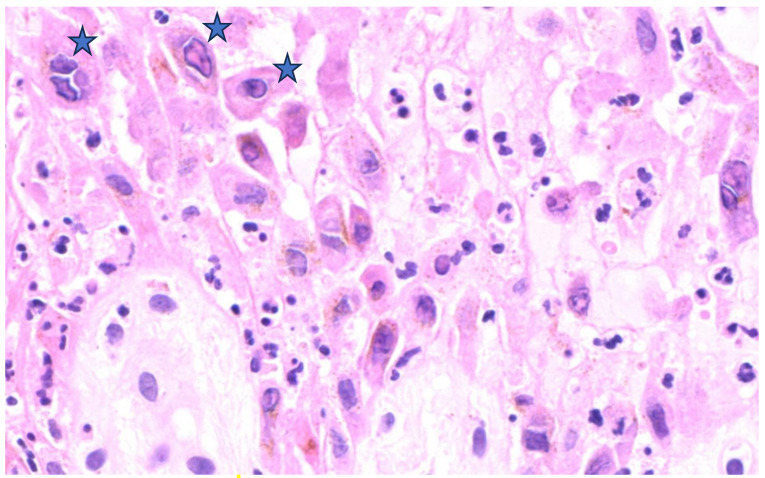
Keratinocytes displaying a marginated nuclear content with a “basophilic halo” around a “ground glass” appearance within the central area of the nucleus (stars). Neutrophilic intraepithelial exocytosis is also evident (H&E, original magnification ×400).

**Figure 6 microorganisms-11-01781-f006:**
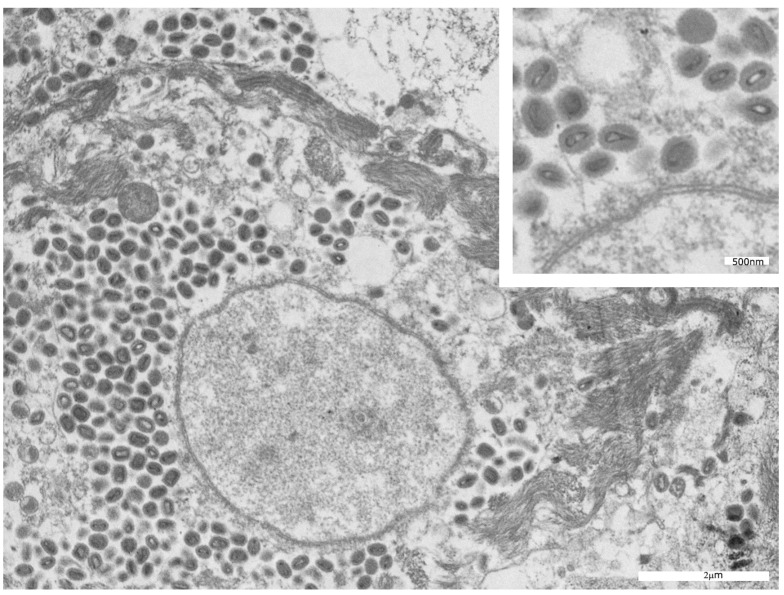
TEM features of MPXV. Mature viral particles, with oval membranous bodies containing biconcave and brick-shaped electron-dense central cores in the cytoplasm of an infected keratinocyte.

**Figure 7 microorganisms-11-01781-f007:**
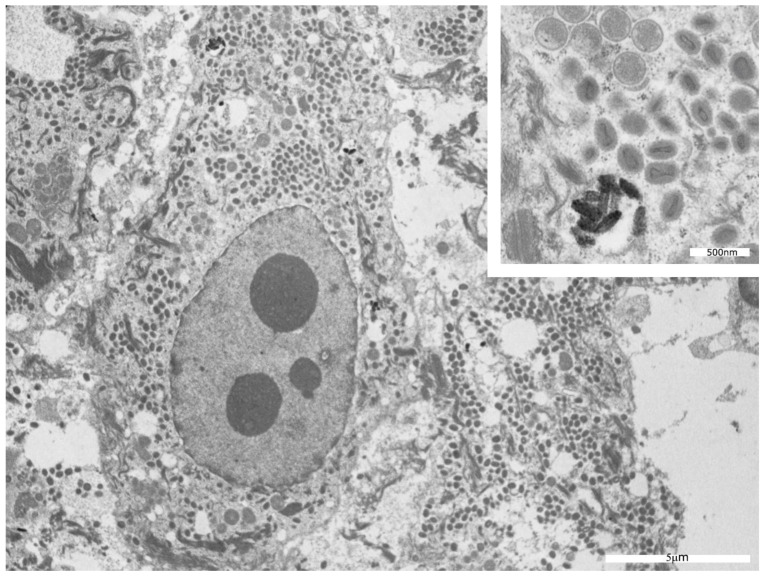
TEM features of MPXV. Roundish particles without any core inside (immature forms) and oval membranous bodies containing biconcave or brick-shaped electron-dense central cores (mature forms) in the cytoplasm of an infected keratinocyte.

**Figure 8 microorganisms-11-01781-f008:**
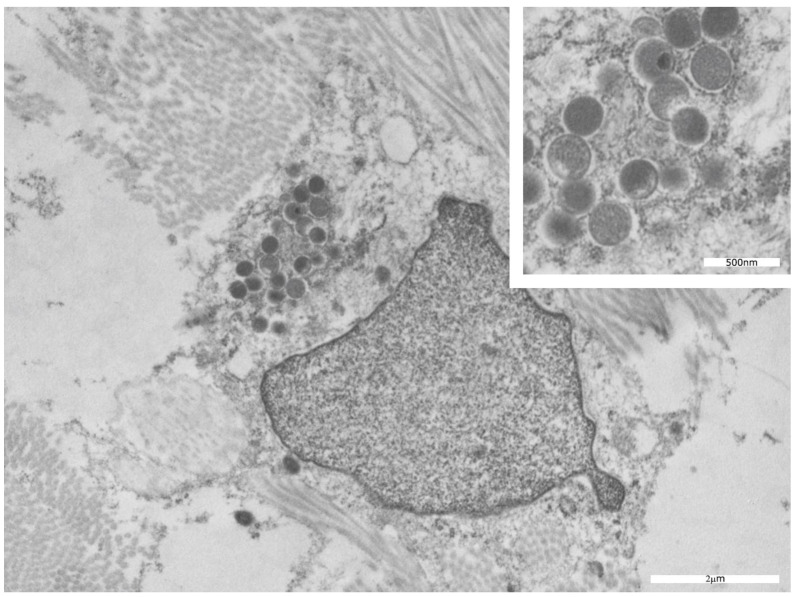
TEM features of MPXV. Mesenchymal cells surrounded by collagen in which many immature viral roundish particles without any core inside were evident. Two mature particles were also observed.

**Table 1 microorganisms-11-01781-t001:** Clinical characteristics of patients with hMPXV infection. MSM: men who have sex with men; STIs: sexually transmitted infections; CR: complete remission.

Pat ID	Sex	Sexual Behavior	Unprotected Sex	Travel History	Other STIs	Kind of Lesions	Number of Skin Lesions	Localization	Systemic Symptoms	Lymphadenopathy	Hospitalization	Outcome
1	M	MSM	Yes	No	None	Pustular lesions	>10	Genital region; lower limb; middle finger of the right hand	Fever; headache; asthenia	No	Yes	CR
2	M	MSM	Yes	Yes (The Netherlands)	None	Pustular lesions	>5	Penile shaft and glans	Fever; headache; myalgia; arthralgia; asthenia	No	No	CR
3	M	MSM	Yes	Yes (Spain)	None	Pustular lesions	<5	Penile shaft and glans; pubic region; perioral region	Asthenia	Yes (laterocervical area)	No	CR
4	M	MSM	Yes	No	Late latent syphilis	Pseudo-pustular lesions	>10	Face; penile shaft; perianal region	Hyperpyrexia; rectal pain	No	No	CR
5	M	MSM	Yes	No	None	Vesico- pustular lesions	>5	Pubic region; penile shaft; upper limb; palm of the hand	Fever; headache; asthenia	No	No	CR
6	M	MSM	Yes	No	Syphilis	Ulcerate lesion	1	Pubic region	Asthenia	Yes (inguinal area)	No	CR

**Table 2 microorganisms-11-01781-t002:** Histopathological features of patients with hMPXV infection.

Pat ID	Site of Biopsy	Histological Stage [14]	Main Cytopathic Modification	Dermal Inflammatory Cell Infiltration
1	Left thigh; penile shaft	Pustular stage	Multinucleated keratinocytes; occasional Guarnieri bodies; extensive ballooning; ground glass nuclei; degenerative modifications in the acrosyringial epithelium	Moderate perivascular and periadnexal with neutrophils
2	Penile	Pustular stage	Ground glass nuclei; ballooning; degenerative modifications in follicular keratinocytes	Mild perivascular lymphocytic infiltration
3	Groin, left shoulder	Pustular stage	Ballooning; occasional Guarnieri bodies; degenerative modifications in follicular keratinocytes	Moderate perivascular and periadnexal with neutrophils
4	Penile shaft	Pustular stage	Guarnieri bodies; ballooning; positive immunohistochemical staining for Treponema pallidum with spirochetes in cytoplasm of keratinocytes	Moderate perivascular and periadnexal with neutrophils
5	Pubic region	Vesicular stage	Ballooning; spongiosis and achantosis; degenerative modifications in the acrosyringial epithelium	Severe perivascular, interstitial, and periadnexal with neutrophils
6	Pubic region	Not applicable	Focal follicular dyskeratosis; positive immunohistochemical staining for Treponema pallidum with perivascular and intraepithelial spirochetes	Severe periadnexal and interstitial lymphocytic infiltration with numerous plasma cells

**Table 3 microorganisms-11-01781-t003:** Electron transmission microscopy features of patients with hMPXV infection.

Pat ID	Lesional Site	Electron Microscopy Transmission Features
1	Left thigh; penile shaft	Almost all keratinocytes contain mature and immature viral particles; virus absent in the stratum corneum and between the scales; virions identified inside the cytoplasm of mesenchymal cells.
2	Penile	Almost all keratinocytes contain mature and immature viral particles; virus absent in the stratum corneum and between the scales.
3	Groin, left shoulder	Almost all keratinocytes contain mature and immature viral particles; virus absent in the stratum corneum and between the scales; virions identified inside the cytoplasm of mesenchymal cells.
4	Penile shaft	Almost all keratinocytes contain mature and immature viral particles; virus absent in the stratum corneum and between the scales.
5	Pubic region	Some viral particles inside the cytoplasm of basal keratinocytes.
6	Pubic region	Absence of the virus.

## Data Availability

All data are present in the main text and in the Appendix A.

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
