# Peer review of "Monkeypox: A Histopathological and Transmission Electron Microscopy Study"

_microorganisms, 2023, doi:10.3390/microorganisms11071781_

Round 1
Reviewer 1 Report
The manuscript provides a comprehensive clinical, histopathological, and transmission electron microscopic description of cutaneous lesions caused by human monkeypox virus infection. The study is well-structured and offers valuable insights into the clinical manifestations and histopathological characteristics of monkeypox virus infection. However, there are several areas where the manuscript could be improved to enhance its clarity, consistency, and overall quality.
Minor Comments:
Abstract: The abstract provides a good overview of the study.
Introduction: The introduction provides a good background on monkeypox virus infection. However, it could be improved by providing more context on the significance of studying the histopathological characteristics of this infection.
Methods: The method’s section is detailed and provides a clear description of the procedures followed in the study. However, the authors should provide further information on the criteria used to select the six patients included in the study.
Results: The results section provides a comprehensive description of the clinical features and histopathological findings of the patients. However, the authors should consider using more visual aids, such as tables or graphs, to present their results more effectively.
References: The authors should ensure that all references are correctly formatted and that all cited studies are included in the reference list.
Major Comments:
Discussion: The discussion section provides a good interpretation of the study's findings. However, it could be improved by providing a more in-depth comparison of the study's findings with those of previous studies. For instance, the authors could discuss how their findings align or contrast with the histopathological characteristics described by Stagles et al. and Bayer-Garner. et al.
And whether these findings are related or contrasted with other Orthopoxviruses.
Ethical Considerations: The authors mention that all procedures were in accordance with the ethical standards of the Helsinki Declaration and that all patients provided written informed consent. However, they should provide more details on the ethical considerations of the study, such as the measures taken to ensure the privacy and confidentiality of the patients.
Limitations: The authors include a section discussing the limitations of their study. For instance, the small sample size of six patients may limit the generalizability of the study's findings.
Conclusion: The manuscript lacks a conclusion section. The authors should include a conclusion that summarizes the main findings of the study, discusses their implications, and suggests directions for future research.
In summary, the manuscript provides valuable insights into the clinical and histopathological characteristics of monkeypox virus infection. However, several improvements could be made to enhance the manuscript's clarity, consistency, and overall quality. The authors should encourage addressing these comments in a revised version of the manuscript.
The quality of English in the manuscript is generally good, with clear and concise language used throughout. However, there are a few areas where improvements could be made to enhance readability and clarity:
1. Sentence Structure: Some sentences in the manuscript are quite long and complex, which can make them difficult to understand. Breaking these down into simpler sentences would enhance readability. For example, in the sentence "In another patient affected by late latent syphilis, immunohistochemical stain for TP showed positive spirochete in the cytoplasm of degenerated keratinocytes, leading to hypothesize that TP colonization could represent the effects of an inflammatory memory recall determined by monkeypox virus on the patient's underlying syphilis", the authors could consider breaking it down into two or three simpler sentences.
2. Grammar and Punctuation: There are a few minor grammatical errors in the manuscript that should be corrected. For instance, ensure consistent use of tenses throughout the manuscript. Also, pay attention to the correct use of articles ("a", "an", "the").
3. Technical Terminology: The manuscript uses a lot of technical terminology, which is appropriate given the subject matter. However, the authors should ensure that all terms are clearly defined when they are first introduced, to make the manuscript accessible to readers who may not be familiar with these terms.
4. Consistency: The authors should ensure consistency in the use of terms throughout the manuscript. For example, if a particular term is abbreviated, the abbreviation should be used consistently throughout the text.
In summary, with minor revisions to sentence structure, grammar, punctuation, and consistency, the quality of English in the manuscript could be significantly improved.
Reviewer 2 Report
Dear Authors,
I appreciate your manuscript and I want to suggest you the follow:
- Line 24 and 112: you wrote "shadow cell". Can you specify and/or add an image to explain this concept?
- Tab. S1: can you specify what STIs is/refer to?
- Line 148: can you rewrite this line for do not use "probably" as concept?
- Line 165: I suggest you to add an image to improve the concept "ground glass"
- Line 192: I suggest you to insert this also into the limitations
Best regards
Round 2
Reviewer 1 Report
I appreciate the effort you have put into addressing the comments and suggestions from the previous review. The revised manuscript has shown significant improvement in terms of clarity, consistency, and overall quality. Here are my comments based on the revised manuscript:
Abstract: The abstract remains clear and concise, providing a good overview of the study.
Introduction: The introduction has been improved by providing more context on the significance of studying the histopathological characteristics of monkeypox virus infection. The additional information enhances the understanding of the study's relevance.
Methods: The methods section remains detailed and provides a clear description of the procedures followed in the study. The additional information on the criteria used to select the patients is appreciated.
Results: The results section has been significantly improved with the addition of visual aids such as tables and figures. This greatly enhances the presentation of the results.
Discussion: The discussion section has been expanded to provide a more in-depth comparison of the study's findings with those of previous studies. The additional comparisons and references enhance the interpretation of the study's findings.
Conclusion: The addition of a conclusion section is appreciated. It effectively summarizes the main findings of the study, discusses their implications, and suggests directions for future research.
Ethical Considerations: The additional details on the ethical considerations of the study, such as the measures taken to ensure the privacy and confidentiality of the patients, are appreciated.
Limitations: The acknowledgment of the small sample size as a limitation of the study remains appropriate.
Quality of English: Based on the revised manuscript, the quality of English language has improved significantly. The authors have addressed the previous concerns about sentence structure, grammar, punctuation, and consistency. The language is clear and concise, and technical terms are well-defined. The manuscript is now more readable and accessible to the audience.
However, I would still recommend a final proofread to catch any minor errors that might have been overlooked. This will ensure that the manuscript is in the best possible form for publication. It's always a good idea to have a native English speaker or a professional language editing service review the manuscript before submission.
Overall, the revised manuscript has addressed the previous comments effectively. The improvements made have enhanced the manuscript's clarity, consistency, and overall quality. I recommend a few minor revisions for further improvement.
Based on the analysis of the revised manuscript, the quality of English language appears to be good. The text is clear and concise, and the technical terminology is used appropriately. However, there are a few areas where minor improvements could be made:
1. Sentence Structure: Some sentences in the manuscript are quite long and complex, which can make them difficult to understand. Breaking these down into simpler sentences would enhance readability.
2. Grammar and Punctuation: There are a few minor grammatical errors in the manuscript that should be corrected. For instance, ensure consistent use of tenses throughout the manuscript. Also, pay attention to the correct use of articles (“a”, “an”, “the”).
3. Technical Terminology: The manuscript uses a lot of technical terminology, which is appropriate given the subject matter. However, the authors should ensure that all terms are clearly defined when they are first introduced, to make the manuscript accessible to readers who may not be familiar with these terms.
4. Consistency: The authors should ensure consistency in the use of terms throughout the manuscript. For example, if a particular term is abbreviated, the abbreviation should be used consistently throughout the text.
In summary, with minor revisions to sentence structure, grammar, punctuation, and consistency, the quality of English in the manuscript could be significantly improved.
Author Response
Thank you very much for your suggestions. We very much appreciated your constructive review.
According to your helpful suggestions, we revised the whole manuscript (sentence structure; grammar and punctuation; technical terminology; consistency and quality of English).